# ABF1 Positively Regulates Rice Chilling Tolerance via Inducing Trehalose Biosynthesis

**DOI:** 10.3390/ijms241311082

**Published:** 2023-07-04

**Authors:** Yazhou Shu, Wensheng Zhang, Liqun Tang, Zhiyong Li, Xinyong Liu, Xixi Liu, Wanning Liu, Guanghao Li, Jiezheng Ying, Jie Huang, Xiaohong Tong, Honghong Hu, Jian Zhang, Yifeng Wang

**Affiliations:** 1State Key Laboratory of Rice Biology and Breeding, China National Rice Research Institute, Hangzhou 311400, China; mm123456m@126.com (Y.S.); liquntang2013@126.com (L.T.); lzhy1418@163.com (Z.L.); liuxinyong1234@gmail.com (X.L.); 18338690086@163.com (X.L.); dearliuwanning@126.com (W.L.); guanghao_12@163.com (G.L.); yingjiezheng@caas.cn (J.Y.); huangjie@caas.cn (J.H.); tongxiaohong@caas.cn (X.T.); 2College of Life Science and Technology, Huazhong Agricultural University, Wuhan 430070, China; huhh@mail.hzau.edu.cn; 3School of Life Sciences, Hubei University, Wuhan 430062, China; zz964419115@163.com

**Keywords:** ABF1, trehalose, chilling tolerance, phosphorylation, rice (*Oryza sativa* L.)

## Abstract

Chilling stress seriously limits grain yield and quality worldwide. However, the genes and the underlying mechanisms that respond to chilling stress remain elusive. This study identified ABF1, a cold-induced transcription factor of the bZIP family. Disruption of *ABF1* impaired chilling tolerance with increased ion leakage and reduced proline contents, while *ABF1* over-expression lines exhibited the opposite tendency, suggesting that ABF1 positively regulated chilling tolerance in rice. Moreover, SnRK2 protein kinase SAPK10 could phosphorylate ABF1, and strengthen the DNA-binding ability of ABF1 to the G-box *cis*-element of the promoter of *TPS2*, a positive regulator of trehalose biosynthesis, consequently elevating the *TPS2* transcription and the endogenous trehalose contents. Meanwhile, applying exogenous trehalose enhanced the chilling tolerance of *abf1* mutant lines. In summary, this study provides a novel pathway ‘SAPK10-ABF1-*TPS2*’ involved in rice chilling tolerance through regulating trehalose homeostasis.

## 1. Introduction

As staple food for more than half of the world’s population, rice (*Oryza sativa* L.) is highly sensitive to chilling stress [1]. With an expanding rice cultivation area from the original tropical and sub-tropical regions to high-latitude areas, chilling stress becomes a major factor that damages crop growth and development, restricts geographical distribution, and eventually constrains yield and quality [2]. Therefore, it is of great value to uncover the molecular mechanism in response to chilling stress for breeding chilling-tolerant varieties.

In the past decades, evidence has shown that trehalose enhances plants’ growth and development in response to cold stress [3,4,5]. The synthesis of trehalose mainly includes two steps, the first is the conversion of UDP (uridine diphosphate)-glucose and glucose-6-phosphate to T6P (Trehalose-6-phosphate) and UDP by TPS (Trehalose-6-phosphate synthase), then TPP (Trehalose-6-phosphate phosphatase) dephosphorylates T6P to trehalose, eventually trehalase (TRE) degraded trehalose into two glucose [6,7]. In Arabidopsis thaliana, over-expression of *TPPI*, a member of the TPP family, improves seedling chilling tolerance through accumulating Jasmonic acid and soluble sugar, such as trehalose, while *tppI* mutant lines are sensitive to low temperatures [8]. Overexpression of *TPS1* or *TPP1* increases trehalose content and strengthens seedlings chilling adaptation in rice [3,9]. Similarly, the over-expression of wheat *TPS11* in Arabidopsis shows a higher survival rate in response to chilling stress compared with the wild type [10]. On the other hand, applying exogenous trehalose could enhance chilling tolerance by increasing endogenous trehalose levels, root growth, and water uptake [3,5,11]. The application of trehalose could also improve the efficiency and quantum yield of PS-II of the photosynthetic system, increase NO and H_2_O_2_ levels to enhance antioxidant activity and suppress membrane lipid peroxidation, whereas reducing the electrolyte leakage to maintain plant growth and harvest in response to cold stress [4].

The basic leucine zipper (bZIP) transcription factors have a bZIP domain composed of two structural features: a DNA-binding base region and a leucine zipper dimerization region [12]. Numerous bZIP transcription factors have been reported to play essential roles in plants’ survival under cold environments [13,14,15]. In rice, the expression of *bZIP52* is strongly induced by low temperatures. Several abiotic adversity-related genes such as *LEA3*, *TPP1,* and *Rab25* are down-regulated in *bZIP52* over-expression (*OxbZIP52*) lines, and *OxbZIP52* shows increased cold sensitivity, suggesting that bZIP52 acts as a negative regulator of cold stress response [16]. bZIP73 regulates rice cold tolerance not only in the seedling period, but also during the heading period. Japonica type bZIP73^Jap^ could interact with bZIP71 to control the balance of abscisic acid (ABA) and reactive oxygen species, thereby improving rice chilling tolerance [13,17]. In wheat, *bZIP60* is induced by ethylene glycol, salt, cold, and ABA, and over-expression of *bZIP60* in Arabidopsis significantly improves the seedling chilling tolerance [18]. bZIP96 is reported to interact with ICE1 (inducer of CBF expression) to positively regulate wheat cold adaptation [19]. In maize, bZIP68, a negative regulator inhibiting seedling chilling tolerance, is phosphorylated by MPK8 to promote protein stability and the DNA binding affinity to the promoter of *DREB 1.7* under chilling stress [14]. In chrysanthemums, Bai et al. (2022) found that bZIP3 could interact with bZIP2 to activate the expression of *POD*, and then promote the peroxidase activity to modulate the balance of reactive oxygen species to improve the cold stress tolerance [15].

The above research shows that both trehalose and bZIP transcription factors are vital for plant chilling tolerance. However, whether a relationship exists between them in response to cold stress is still uncovered. This study identified that ABF1 (bZIP12) acts as a positive regulator in rice chilling tolerance by inducing trehalose biosynthesis through the novel pathway ‘SAPK10-ABF1-*TPS2*’. Briefly, *abf1* mutant lines exhibited increased chilling sensitivity by interfering with trehalose biosynthesis. SnRK2 protein kinase SAPK10 could phosphorylate ABF1 and promote the latter’s binding ability to *TPS2*, a member of the trehalose-6-phosphate synthase family, ultimately inducing trehalose biosynthesis under chilling stress.

## 2. Results and Discussion

### 2.1. ABF1 Acts as a Positive Regulator of Rice Chilling Tolerance

In this study, we first analyzed the expression level of *ABF1* in response to cold treatment at 0, 1, 3, 6 and 12 h. Upon cold treatment, the transcription level of *ABF1* sharply rose to about 3 fold at 1 h, and kept steady until 6 h, then dropped slowly to about 2.5 fold at 12 h, which indicated that ABF1 might participate in rice chilling adaption (Appendix A). To explore the biological role of ABF1, two independent homozygous mutant lines (*abf1-1* and *abf1-2*) and two *ABF1* over-expression lines (*OxABF1-1* and *OxABF1-9*) were generated (Appendix A). The *abf1-1* and *abf1-2* mutant lines harbored a TC deletion and a TCGC deletion in the first exon and shifted the open reading frame of *ABF1*, respectively (Appendix A). As shown in Figure 1, under normal conditions, no significant difference in seedling growth was distinguished between the wild-type and mutant lines (Figure 1). However, the leaves of the mutant lines showed more severe wilting and crimping than the wild-type under cold treatment for 5 days after 7-days recovery, with the survival rates (less than 23%) notably lower than that of the wild type (average 32%), while the survival rates of *ABF1* over-expression lines (average 49%) were significantly higher than the wild type (Figure 1 and Figure 2A,B). Previous reports have shown that chilling stress affects numerous physiological and cellular processes in plants, including inhibiting photosynthesis, regulating calcium signals, and altering the organelle’s ultrastructure, especially the chloroplast lipid membrane [20]. Accordingly, plants employ several strategies when recovering from chilling to normal temperatures, such as inducing the expression of photosynthetic enzymes, enhancing the linoleic acid metabolism and inhibiting phenylpropanoid biosynthesis and ribosome synthesis [21]. Among these processes, maintaining the cell membrane integrity and stability is vital for chilling tolerance, and ion leakage reflects the cell membrane’s damage degree, while proline declines the electrolyte leakage level to protect membrane stability [22,23]. In accordance with the increased cold sensitivity of *abf1*, the mutant lines’ electrolyte leakage was higher than the wild type, accompanied by reduced proline contents (Figure 2B–D). In contrast, *ABF1* over-expression lines exhibited the opposite tendency under cold stress (Figure 2B–D). These results demonstrated that ABF1 positively participated in rice chilling tolerance.

Previous reports have shown that ABF1 is involved in various abiotic stress responses, such as hypoxia, salt, drought, and cold adversity stress [24]. *abf1* mutant lines are hypersensitive to drought and salt stresses, and *ABF1* over-expression lines significantly improve drought tolerance [25,26]. Moreover, ABF1 acts as a negative regulator for flowering transition and is functionally redundant with bZIP40, which may delay heading by directly activating *WRKY104* and indirectly inhibiting *Ehd1* under drought conditions [26]. Our lab recently reported that ABF1 regulates plant height and seed germination by inhibiting gibberellin synthesis in rice [27]. Overexpression of *ABF1* results in a typical gibberellin deficient phenotype, exhibiting semi-dwarfism and retarded seed germination, which could be restored by applying exogenous GA_3_ [27]. In this study, we have detected that *ABF1* was induced by chilling stress, which was consistent with the described previously, and further found a novel role of ABF1 involved in elevating rice chilling adaption, indicating ABF1 could play pleiotropic roles in regulating various abiotic stresses tolerance.

### 2.2. SAPK10 Directly Interacts with and Phosphorylates ABF1

After screening a rice seedling-derived cDNA library using a yeast two-hybrid system, we found that SAPK10, a member of the specific Ser/Thr SnRK2 (Sucrose non-fermenting-1-related protein kinase 2) type kinase family, could interact with ABF1 (Figure 3A). Intriguingly, the transcription level of *SAPK10* was also significantly elevated under cold treatment (Appendix A). A bimolecular fluorescence complementation (BiFC) assay detected no YFP fluorescence signals when YN-ABF1 or YC-SAPK10 was expressed alone, and ABF1 and SAPK10 could interact to generate YFP fluorescence in the nuclei of tobacco leaf cells (Figure 4). In the GST pull-down assay, the purified His-SAPK10 was pulled down with GST-ABF1, but not with GST, and Coimmunoprecipitation (Co-IP) assay exhibited a similar result, ABF1-GFP was immunoprecipitated by SAPK10-FLAG, while GFP was not (Figure 3B,C). Moreover, in vitro kinase assays detected a phosphorylation band of purified GST-ABF1 (p-GST-ABF1) when GST-ABF1 co-expressed with His-SAPK10 in *E. coli*, and the relative strength of p-GST-ABF1 band was reduced about 23% of that of GST-ABF1 band after CIAP treatment (Figure 3D). Noteworthy, the band shift mobility of p-GST-ABF1 is significantly slower than that of GST-ABF1, further supporting the highly specific of the SAPK10-mediated ABF1 phosphorylation (Figure 3D). Additionally, we also detected the auto-phosphorylation band of SAPK10-FLAG, which was consistent with previous work [28].

### 2.3. Disruption of SAPK10 Impairs Rice Chilling Tolerance

To further clarify the role of SAPK10 in regulating rice chilling tolerance, two independent homozygous mutant lines (*sapk10-2* and *sapk10-3*) and two *SAPK10* overexpression lines (*OxSAPK10-1* and *OxSAPK10-3*) were used for further analysis, respectively (Appendix A). The *sapk10-2* and *sapk10-3* mutant lines harbored an A insertion or an A/G insertion in the first exon and shifted the open reading frame of *SAPK10* (Appendix A). Interestingly, *sapk10* mutant lines displayed a phenotype similar to *abf1* under cold stress. *sapk10* mutant lines also exhibited more severe wilting and crimping than the wild type, with the survival rates (less than 25%) lower than the wild type (Figure 5 and Figure 6A). Although the height of *SAPK10* overexpression lines was shorter than the wild type under normal conditions as previous report [28], the survival rates (average 40%) of *SAPK10* overexpression lines were higher compared with the wild type under cold treatment (Figure 5 and Figure 6B). In accordance with the above observations, the electrolyte leakage of sapk10 increased more rapidly than in the wild type. In contrast, the electrolyte leakage was reduced in *SAPK10* overexpression lines under cold treatment, whereas the proline contents of these transgenic lines showed an opposite tendency (Figure 6C,D). These experiments showed that SAPK10 enhanced rice chilling adaptation.

SnRK2 is a specific Ser/Thr protein kinase that plays essential roles in osmotic stress response and other abiotic stress processes [29]. In rice, there are ten SnRK2 family members, namely SAPK1-SAPK10, of which SAPK6 and SAPK8 act as crucial protein kinases in the low-temperature signal transduction pathway [30,31]. Under cold stress, SAPK6 interacts with and phosphorylates IPA1 at Serine 201 and 213 sites to stabilize the protein stability of IPA1, thereafter enhancing IPA1 accumulation and activating the expression level of *CBF3*, ultimately enhancing rice cold stress adaption [31]. Wang et al. (2021) report that SAPK8 phosphorylates a cyclic nucleotide-gated channel CNGC9, activating its channel activity, and promoting cold-induced calcium influx and cytoplasmic calcium elevation in rice [30]. In this study, we found a novel role of SAPK10 involved in positively regulating rice chilling tolerance, though it has been reported to either phosphorylate bZIP72 or bZIP20 to participate in abscisic acid and jasmonic acid synergistically inhibiting seed germination or enhancing drought and salt tolerance processes [28,32]. Interestingly, SAPK6, SAPK8, SAPK10, and ABF1 are all induced by abscisic acid (ABA) and act as positive regulators in the ABA signal pathway [25,29,33]. Since ABA biosynthesis is strengthened during cold stress and a series of ABA-responsive genes are induced under cold treatment, whereas the mutant lines interfering with ABA biosynthesis exhibit impaired cold adaptation, and application of exogenous ABA enhances chilling tolerance, suggesting that ABA functions as a vital phytohormone positively regulating cold stress response [34,35]. Whether the three protein kinases or other members of the SnRK2 family interact with ABF1 involved in the crosstalk between ABA and chilling tolerance would be an interesting topic to be further investigated.

### 2.4. abf1 Inhibits Seedling Cold Adaptation by Interfering with Trehalose Homeostasis

Previous reports have shown that trehalose improves chilling tolerance in several species [3,8,36], which triggered us to speculate that ABF1-mediated chilling alleviation may be partially based on trehalose accumulation. We first detect the endogenous trehalose level in *abf1* mutant lines to verify this hypothesis. As anticipated, the *abf1* seedlings had lower trehalose levels than the wild-type under cold treatment, while there was no difference under normal conditions (Figure 7A). Furthermore, applying exogenous trehalose could enhance the chilling tolerance of *abf1*, relieving severe wilting and crimping phenomena (Figure 8). Notably, the relative survival rates of *abf1* were significantly higher than the wild type when exogenous trehalose was added (Figure 7B,C). Accordingly, the endogenous trehalose contents in *ABF1* over-expression lines were higher than the wild-type in response to chilling stress, though slightly higher than the wild-type under normal conditions (Figure 7D). Thus, these results strongly suggested that ABF1 promotes trehalose contents to enhance seedlings chilling tolerance.

The underlying mechanisms of bZIP transcription factors regulating rice cold tolerance have been reported in the past decades. Liu et al. (2018) have shown that one single nucleotide polymorphism difference in the coding region of *bZIP73* causes the difference in low-temperature tolerance between Indica and Japonica rice. And Japonica type bZIP73^Jap^ interacts with bZIP71 to regulate abscisic acid (ABA) level and active oxygen balance, thus improving rice tolerance to low temperatures [13]. Moreover, the regulatory effect of bZIP73^Jap^ on low-temperature tolerance also occurs during the booting stage. Under natural cold stress conditions, the co-expression of *bZIP73^Jap^* and *bZIP71* transgenic lines significantly improves seed setting rate and grain yield, not only inhibiting ABA levels in anthers, but also promoting the transportation of soluble sugars from anthers to pollen [17]. This study found that trehalose, except ABA, active oxygen and soluble sugars, also participates in bZIPs-mediated cold tolerance processes. Under low-temperature stress, applying trehalose increases NO levels and stimulates H_2_O_2_ contents, resulting in enhanced antioxidant activity and reduced membrane lipid peroxide activity [4]. In addition, exogenous trehalose treatment increases the contents of endogenous spermidine, glutathione, and ascorbic acid, promotes the glutathione and ascorbic acid cycle, and finally reduces the ROS production to improve the cold tolerance of plants [36]. Hence, a possibility exists that trehalose may play a role in ABF1-mediated low-temperature tolerance by affecting the active oxygen homeostasis in vivo.

### 2.5. ABF1 Binds to and Activates TPS2 Expression

Since ABF1 strengthens rice seedling adaptation by inducing trehalose accumulation, indicating that ABF1 may induce trehalose biosynthesis or suppress trehalose catabolism. This hypothesis triggered us to detect the expression level of a series of crucial enzymes, including TPS (Trehalose-6-phosphate synthase), TPP (Trehalose-6-phosphate phosphatase), and TRE (Trehalase) involved in the trehalose biosynthesis and catabolism pathway [6,7], in *abf1* seedlings under cold treatment. As shown in Figure 9A, *TPS2*, *TPP5* and *TPP6* involved in trehalose biosynthesis were significantly reduced in *abf1* mutant lines (Figure 9A). Among these enzymes, TPS2 attracted our particular interest due to ABF1 binds the G-box (CACGTG) motif of *TPS2* by EMSA (electrophoresis mobility shift assay), and the binding ability disappeared after the unlabeled probe was added (Figure 9C). To our surprise, ABF1 did not interact with other *TPP* or *TPS* members, of which the promoters also harbored G-box cis-elements predicted by PlanCARE (Plant Cis-Acting Regulatory Elements) [37] (Appendix A). Yeast-one hybrid (Y1H) assay further confirmed that ABF1 could bind the promoter of *TPS2* and activate the LacZ expression, though it displayed weak autoactivation activity (Figure 9B). The binding pattern of ABF1 on the promoter of *TPS2* was further validated by ChIP-qPCR (Chromatin immunoprecipitation-quantitative PCR) under chilling stress. ABF1 was notably enriched in the P3 region containing the G-box motif of the promoter of *TPS2*, nearly threefold higher than the other probes (Figure 9D). Finally, a luciferase transient transcriptional activity assay (LUC) was used to detect the regulatory effect of ABF1 on *TPS2*. The luciferase activity of the *proTPS2*: firefly LUC reporter was significantly facilitated by *pro35S*: ABF1: tNOS, which was consistent with the expression pattern of *TPS2* in *abf1* (Figure 9A,E). Intriguingly, the upregulation of *TPS2* by ABF1 was remarkably enhanced when *pro35S*: SAPK10: tNOS was added, indicating that SAPK10-mediated phosphorylation on ABF1 promotes ABF1-mediated *TPS2* expression (Figure 9E). On the other hand, *TPS2* was also induced in response to cold treatment, which was similar to the expression pattern of ABF1 and SAPK10 (Appendix A). These results strongly implied that ABF1 specifically binds to *TPS2* and activates the latter’s expression.

Moreover, we tested the chilling tolerance of *tps2* mutant lines and *TPS2* overexpression lines (*OxTPS2-1* and *OxTPS2-2*) (Appendix A). *tps2* mutant lines harbored a T deletion and a T insertion in the first exon, and shifted the open reading frame of *TPS2*, respectively (Appendix A). Moreover, the trehalose contents were substantially reduced in *tps2* mutant lines, and increased in *TPS2* overexpression lines (Appendix A). In accordance with this, *tps2* mutant lines displayed more severe wilting and crimping, and lower survival rates than the wild type, while *TPS2* overexpression lines exhibited the opposite tendency (Figure 10 and Figure 11A,B). Accordingly, the electrolyte leakage of *tps2* was higher, accompanied by reduced proline content, while *TPS2* overexpression lines showed lower electrolyte leakage and higher proline concentration compared with the wild type in response to low-temperature stress (Figure 11C,D). These findings demonstrated that TPS2 positively participates in rice seedling chilling tolerance, which was up-regulated by ABF1 to enhance trehalose biosynthesis.

As a non-reducing sugar found in bacteria and yeasts, trehalose serves as the source of carbon in higher plants. Trehalose regulates a series of physiological, biochemical, and molecular mechanisms in higher plants. Trehalose supplementation promotes root growth and water absorption, and thus improves plant growth under cold stress [11]. Trehalose supplementation also improves the efficiency of PS-II and the quantum yield of PS-II, while reducing the leakage of electrolytes to adapt to chilling stress [4]. On the other hand, trehalose stimulates the antioxidant defense system, such as CAT, APX, to maintain cellular integrity under cold treatment [38]. The functions and the underlying mechanisms of several TPS or TPP members involved in rice chilling tolerance have been reported [39]. TPS1 enhances rice’s abiotic tolerance by increasing the trehalose and proline content and inducing the expression of stress-related genes [9]. Overexpression of *TPS1* increased the tolerance of transgenic rice seedlings to low temperature, salt, and drought treatments, with the concentrations of trehalose and proline higher than the wild type [9].

MAPK3-bHLH002-*TPP1* pathway positively regulates cold tolerance in rice. MAPK3 kinase phosphorylates bHLH002/ICE1 and inhibits the latter’s ubiquitination by HOS1, then bHLH002 activates *TPP1* expression and ultimately enhances rice cold tolerance [3]. Recently, Wang et al. (2021) further report that TPP1 controls rice seed germination through crosstalk with the abscisic acid (ABA) catabolic pathway [40]. *tpp1* mutant lines down-regulate ABA catabolism and inhibit seed germination. The transcription factor GAMYB can activate the expression of *TPP1*, increase the trehalose content and induce the transcriptional abundance of the ABA catabolic genes to decrease ABA content in seeds [40]. In this study, we first report the function of cold-induced *TPS2* in strengthening seedling cold adaption by enhancing trehalose and proline contents, similar to TPS1. A previous study has reported auxin and cytokinin regulate root system architecture under low-temperature stress, mainly through modulating root cell division, differentiation, and elongation. Moreover, cytokinin determines root architecture and function by fine-turning auxin transport and signaling [41], indicating the cross-talk of different phytohormones controlling the stem cells of the root in response to chilling stress. Recently, our lab found that ABA-induced SAPK10 can phosphorylate bZIP72 and enhance the latter’s binding ability to the promoter of *AOC*, an essential gene in the jasmonic acid (JA) synthesis pathway, thereby inducing *AOC* expression and increasing endogenous JA levels, finally inhibiting seed germination [28]. Based on these regulators were all induced by cold treatment, the reduced trehalose and proline contents in the mutant lines under chilling conditions, and the interaction among the three regulators, we established a novel pathway ‘SAPK10-ABF1-*TPS2*’ in trehalose-mediated rice cold tolerance in this study (Figure 12). Since ABA and trehalose enhance cold stress tolerance, SAPK10 and ABF1 are ABA signalling positive regulators, and TPS2 is the key enzyme involved in trehalose biosynthesis [25,29,35,39]. Further study will focus on the relationship between trehalose and ABA homeostasis in response to chilling stress, which would deepen our understanding of the regulatory mechanism acclimating to low temperatures.

## 3. Materials and Methods

### 3.1. Vector Construction and Plant Transformation

For overexpression of *ABF1*, *SAPK10* and *TPS2*, the coding sequence (CDS) was fused into the binary vector PU1301 driven by the maize ubiquitin promoter using the *Kpn*I and *Bcl*I sites, respectively. *abf1*, *sapk10* and *tps2* mutant lines were generated by CRISPR/Cas9 system as described previously [42]. Briefly, annealed double-strand oligos 20 nt in the length of the gDNA sequence were ligated into the pYLCRISPR/Cas9-MH vector. All the vectors were transformed into Nipponbare (*Oryza sativa* L. ssp. *japonica* cv Nipponbare) callus using the *Agrobacterium*-mediated transformation method [43]. The primers used were listed in Appendix A.

### 3.2. Chilling Stress Treatment

Chilling stress treatment at the seedling stage was conducted as described previously [44] with a few modifications. Briefly, 7-day-old rice seedlings grown using rice nutrient solution [45] were transferred to a growth chamber at 4 °C (14 h: 10 h, light: dark, approximately 200 µmol m^−2^ s^−1^ photon density) for 5 days. Then the seedlings were restored for 7 days in a growth chamber at 28 ± 2 °C (14 h: 10 h light: dark, approximately 200 µmol m^−2^ s^−1^ photon density). For the short-time chilling stress treatment, samples were taken at 0, 1, 3, 6.12 h (light, 4 °C), respectively.

### 3.3. Yeast Hybrid Assay

Yeast two-hybrid assay was performed according to Matchmaker™ Gold Yeast Two-Hybrid System (Clontech, CA, USA). The CDS of *ABF1* was cloned into the pGADT7 plasmid at the *EcoR*I and *Pst*I sites. The CDS of *SAPK10* was ligated to the pGBKT7 vector at the *EcoR*I and *BamH*I sites. The recombinant vectors were co-transformed into the yeast strain Y2H Gold (Clontech) and selected on a synthetic medium lacking leucine, tryptophan, histidine and Adenine with 0.04 mg mL^−1^ X-α-Gal and 100 ng mL^−1^ Aureobasidin A added.

Yeast one-hybrid assay was performed according to Matchmaker™ One-Hybrid System (Clontech, CA, USA). The CDS of *ABF1* was cloned into the pB42AD plasmid at the *EcoR*I and *Xho*I sites. The 2 kb promoter of *TPS2* was ligated to the pLacZ2μ vector at the *Kpn*I and *Xho*I sites. The recombinant vectors were co-transformed into the yeast strain EGY48 and selected on a synthetic medium lacking tryptophan and uracil with 1% raffinose, 1× BU salts, 80 mg/L X-Gal, and 2% galactose added. The primers used were listed in Appendix A.

### 3.4. BiFC Assay

For Bimolecular Fluorescence Complementation (BiFC) assay, the CDS of *ABF1* was cloned into pDOE plasmid at the *BspE*I site, and the CDS of *SAPK10* was fused to the pDOE at the *BamH*I site as described previously [46]. The recombinant vectors were then transformed into *Agrobacterium* strain EHA105, and infiltrated into tobacco leaves. Fluorescence signals were observed using a Zeiss LSM710 confocal laser-scanning microscope (Carl Zeiss AG, Jena, Germany) after 72 h of infiltration. The primers used were listed in Appendix A.

### 3.5. In Vitro Pull-Down Assay

The CDS of *ABF1* and *SAPK10* were fused into pGEX-4T-1 (G.E. Healthcare, Chicago, IL, USA) and pET-28a (Thermo, Waltham, MA, USA) vectors, respectively. The GST-ABF1, GST and His-SAPK10 were transformed into *Transetta* (DE3) chemically competent cells (Transgen, Beijing, China), and purified using the glutathione S-transferase (GST)-Sefinose^TM^ Kit (Sangon Biotech, Shanghai, China) and 6× His-Tagged Protein Purification Kit (CWBIO, Beijing, China), respectively. The purified proteins were incubated with 50 µL Glutathione High Capacity Magnetic Agarose Beads (Sigma-Aldrich, St. Louis, MO, USA) and 600 µL pull-down buffer (50 mM Tris-HCl, pH 7.5, 5% glycerol, 1 mM EDTA, 1 mM Dithiothreitol (DTT), 1 mM phenylmethylsulfonyl fluoride (PMSF), 0.01% NonidetP-40, and 150 mM KCl) at 4 °C for 2 h. Then the beads were washed five times with pull-down buffer, and suspended in 50 μL of 1× PBS and 10 μL of 6× SDS protein loading buffer, boiled for 5 min, and resolved on 10% acrylamide gels. Individual bands were detected using Supersignal West Pico Chemiluminescent Substrate (Thermo, Waltham, MA, USA) and the ChemDoc^TM^ Touch Imaging system (Bio-Rad, Hercules, CA, USA). The dilution for anti-GST (Yeasen, Shanghai, China) and anti-His (Yeasen, Shanghai, China) was 1: 5000. The primers used were listed in Appendix A.

### 3.6. Coimmunoprecipitation (Co-IP) Assays 

The CDS of *SAPK10* was digested with *BamH*I and cloned into the *proUbi*–FLAG vector, and the CDS of *ABF1* was digested with *Kpn*I and *Xba*I and cloned into pCAMBIA1300-35S-GFP vector, respectively. SAPK10-FLAG was transiently co-expressed with an empty green fluorescent protein (GFP) or ABF1-GFP in tobacco leaves by *Agrobacterium* infiltration. The total protein of cotransformed tobacco leaves was extracted by extraction buffer (25 mM Tris-HCl, pH 7.4, 150 mM NaCl,1 mM EDTA, 1% NonidetP-40, 5% glycerol, 1 mM PMSF, 20 μM MG132 and 1× Roche protease inhibitor cocktail (Roche, Basel, Switzerland)). After centrifugation twice (12,000× *g* for 10 min each time), the supernatant was incubated with anti-FLAG M2 magnetic beads (Sigma-Aldrich, St. Louis, MO, USA) at 4 °C for 2 h. Then the beads were washed five times with washing buffer (50 mM Tris, pH 7.5, 150 mM NaCl, 0.2% Triton X-100, 1 mM PMSF, and 1× Roche protease inhibitor cocktail). The immunoprecipitated proteins were suspended in 50 μL of 1× PBS and 10 μL of 6× SDS protein loading buffer, boiled for 5 min, and resolved on 10% acrylamide gels. Individual bands were detected using Supersignal West Pico Chemiluminescent Substrate and the ChemDoc^TM^ Touch Imaging system. The dilution for anti-FLAG (Sigma-Aldrich, St. Louis, MO, USA) and anti-GFP (Yeasen, Shanghai, China) antibodies was 1: 5000. The primers used were listed in Appendix A.

### 3.7. In Vitro Kinase Assay

In vitro kinase assay was performed as described previously [28]. Briefly, His-SAPK10 was co-expressed with GST-ABF1 in *Transetta* (DE3) chemically competent cell (Transgen, Beijing, China), and purified using the GST-Sefinose^TM^ Kit (Sangon Biotech, Shanghai, China) and 6× His-Tagged Protein Purification Kit (CWBIO, Beijing, China), respectively. The purified proteins (100 ng) were incubated with 1 µg calf intestinal phosphatase (CIAP; Takara, Dalian, China) at 37 °C for 30 min, and separated by electrophoresis on 10% acrylamide gels. Phosphorylated bands were detected using biotinylated Phos-tag^TM^ zinc complex BTL111 from Wako (Osaka, Japan). The protein intensities were quantified by ImageJ 1.47V software (ImageJ bundled with 32-bit Java 1.6.0_20).

### 3.8. Electrophoretic Mobility Shift Assay (EMSA)

GST-ABF1 was expressed in *Transetta* (DE3) chemically competent cell (Transgen, Beijing, China), and purified using the GST-Sefinose^TM^ Kit (Sangon Biotech, Shanghai, China). The probes in a length of 59 bp in the *TPS2* promoter were commercially synthesized by Tsingke (Tsingke Biotech, Hangzhou, China) and labeled by an EMSA Probe Biotin Labeling Kit (Beyotime, Shanghai, China). Briefly, an equal amount of purified recombinant proteins was pre-incubated with EMSA/gel-shift binding buffer, and then incubated with 20 fmol labeled probes or nonlabeled competitive DNA probes. Then the incubated samples were separated by electrophoresis on 6% acrylamide gels. The labeled DNA probes were detected by the LightShift Chemiluminescent EMSA Kit (Thermo, Waltham, MA, USA). The primers used were listed in Appendix A.

### 3.9. Luciferase Transient Transcriptional Activity Assay (LUC)

The CDS of *ABF1* and *SAPK10* were fused into ‘None’ vectors as effectors by using the *BamH*I and *EcoR*I sites, and the 2 kb promoter of *TPS2* was ligated into 190fLUC vector as a reporter by using the *Hind*III site, respectively. The plasmids were co-transformed into rice protoplasts as described previously [47]. The luciferase activities were detected by the Dual-Luciferase Reporter Gene Assay Kit (Beyotime, Shanghai, China) using the Promega GLOMAX system (Promega, Madison, WI, USA) according to the manufacturer’s instructions. The relative luciferase activity was defined as the ratio between fLUC and rLUC (fLUC/rLUC). *AtUbi3*:rLUC was adopted as the internal control. The primers used were listed in Appendix A.

### 3.10. Chromatin Immunoprecipitation-Quantitative PCR (ChIP-qPCR)

The ChIP-qPCR was carried out as described previously [48]. Briefly, chromatin was isolated from 3 g crosslinked leaves of the wild-type under chilling stress at 4 °C for 12 h. Isolated chromatin was sonicated for DNA fragmentation ranging from 200 to 700 bp. Subsequently, sonicated DNA/protein complexes were immune-precipitated with polyclonal rabbit ABF1 antibody against the amino acid residues 149 to 162 (C-KGQEEAPDGSDGPR-COOH) of ABF1, which was commercially synthesized and affinity-purified by Genescript (Genescript, Shanghai, China). After reverse cross-linking and proteinase K treatment, the immunoprecipitated DNA was purified with phenol/chloroform. Then the immunoprecipitated and input DNA were used for quantitative polymerase chain reaction (qPCR), respectively. The primers used were listed in Appendix A.

### 3.11. Trehalose Extraction and Quantification

Briefly, the leaves of 7-day-old seedlings (0.1 g) before or after chilling stress at 4 °C for 72 h were homogenized in 5 mL of 80% (*v*/*v*) hot ethanol for 20 min and centrifuged. Then the supernatants were collected for trehalose extraction and quantification using the Trehalose Content Kit (Grace Biotechnology, Suzhou, China) according to the manufacturer’s instructions. The absorbance of trehalose was measured at 620 nm.

### 3.12. Proline Determination

Briefly, 0.1 g leaves of 7-day-old seedlings before or after chilling stress at 4 °C for 5 days were immersed in 5 mL 3% (*v*/*v*) hot sulfosalicylic acid solution for 30 min, during which the samples were constantly shaken and cooled to room temperature to prepare crude proline extract. A total of 1 mL crude proline extract was added to 2 mL reaction buffer containing 1 mL of 2.5% acidic ninhydrin (60 mL glacial acetic acid, 16.4 mL phosphoric acid, 36.6 mL deionized water, and 2.5 g ninhydrin) and 1 mL glacial acetic acid, then boiled for 30 min with continuous shaking before cooling to room temperature. Finally, 0.5 mL toluene was used to extract 0.75 mL reaction solution with vortexing for 1 min, then incubated for 30 min. The absorbance of proline was measured at 520 nm by a SpectraMax i3x Multi-Mode Microplate Reader (MOLECULAR DEVICES, Sunnyvate, CA, USA).

### 3.13. Membrane Ion Leakage Measurement

Membrane Ion Leakage was determined as described previously with a few modifications [49]. Leaves of 7-day-old seedlings weighed at 0.3 g before or after chilling stress at 4 °C for 5 days were immersed in 30 mL deionized water and shaken at 120 rpm for 4 h. The membrane ion leakage L_1_ was measured using a DDS-307A conductivity meter (LeiCi, Hangzhou, China). Then, the leave samples were incubated in boiling water for 30 min. After cooling down to room temperature, the membrane ion leakage L_2_ of the corresponding leaves was measured. The formula EL = L_1_/L_2_ × 100% was used to calculate the ion leakage of leaves.

## 4. Conclusions

In this study, we characterized the function of ABF1 positively participating in rice chilling tolerance. *ABF1* over-expression lines strengthened chilling tolerance with reduced ion leakage and increased proline contents, while *abf1* mutant exhibited the opposite tendency. Furthermore, SnRK2 protein kinase SAPK10 directly interacts with and phosphorylates ABF1, and strengthens the DNA-binding ability of ABF1 to a member of Trehalose-6-phosphate synthase *TPS2*, thereby facilitating the transcription of *TPS2* and endogenous trehalose biosynthesis under chilling stress. In a word, this study revealed a novel chilling tolerance strategy mediated by ‘SAPK10-ABF1-TPS2’ through fine-turning trehalose homeostasis.

## Figures and Tables

**Figure 1 ijms-24-11082-f001:**
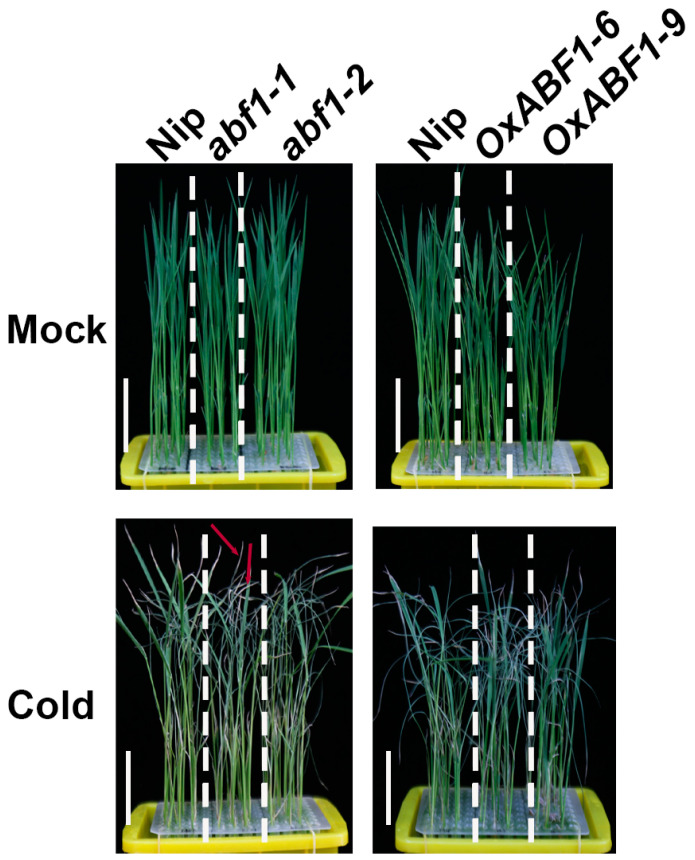
ABF1 positively regulates rice seedling chilling tolerance. *abf1* mutant lines are less tolerant to chilling stress with more severe wilting and crimping leaves than the wild type. *ABF1* overexpression lines showed the opposite tendency. Seven-day-old seedlings were hydroponically cultured under chilling stress (4 °C) for 5 days and photographed. Red arrow indicated the wilting and crimping leaves. Bar = 3 cm.

**Figure 2 ijms-24-11082-f002:**
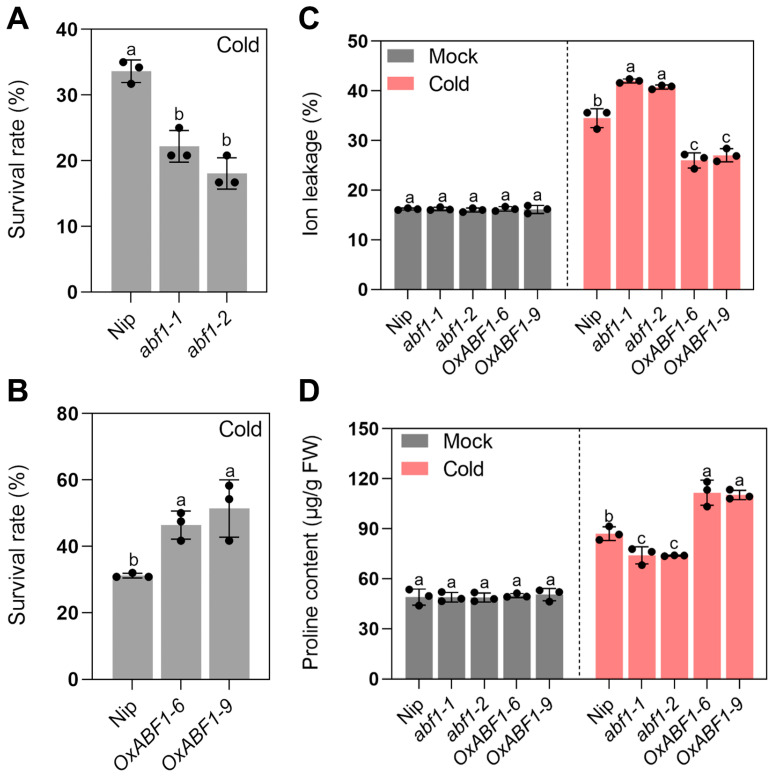
(**A**) Survival rates of wild-type, *abf1* mutant lines under chilling stress (4 °C) for 5 days. Survival rates (%) = the number of seedlings that survived after chilling stress treatment/total number of the seedlings (%). (**B**) Survival rates of wild type, *ABF1* overexpression lines under chilling stress (4 °C) for 5 days. (**C**) Ion leakage of seedling leaves from wild type, *abf1* mutant lines, and *ABF1* overexpression lines under chilling stress (4 °C) for 5 days. (**D**) Proline contents of seedling leaves from wild type, *abf1* mutant lines and *ABF1* overexpression lines under chilling stress (4 °C) for 5 days. Error bars indicate SD with biological triplicates (*n* = 3). Tukey’s test with one-way analysis of variance (ANOVA). Different letters indicate statistical differences at *p* < 0.05.

**Figure 3 ijms-24-11082-f003:**
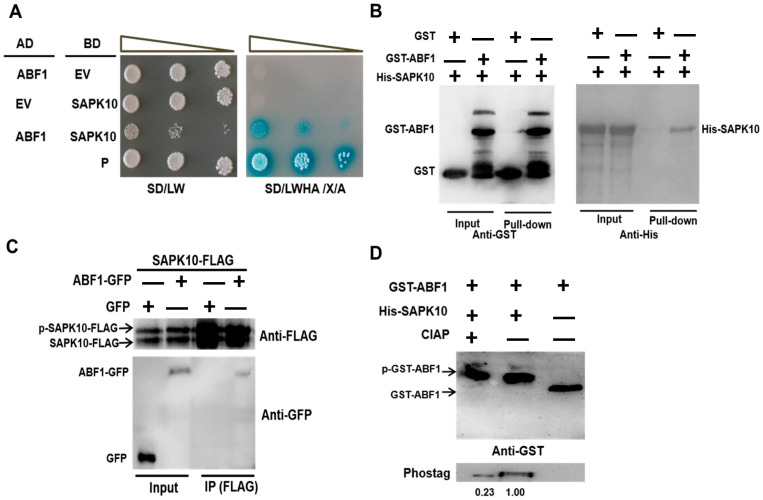
SAPK10 physically binds to and phosphorylates ABF1. (**A**) Yeast two-hybrid assays. Yeast cells cotransformed with SAPK10-AD and ABF1 fused to the GAL4-binding domain (ABF1-BD) were grown on selective media. BD, pGBKT7; AD, pGADT7; EV, empty vector; SD/LW, -Leu-Trp; SD/LWHA, -Leu-Trp-His-Adenine; P, positive control using pGADT7-T + pGBKT7-53; X: X-a-Gal in 0.04 mg mL^−1^; A: Aureobasidin A in 100 ng mL^−1^. (**B**) Pull-down assay. Purified His-SAPK10, GST and GST-ABF1 were subjected to pull-down assays and detected with anti-His and anti-GSTantibodies, respectively. (**C**) Coimmunoprecipitation (Co-IP) assay. GFP, ABF1-GFP and SAPK10-FLAG extracted from infiltrated tobacco (Nicotiana benthamiana) leaves were used in a Co-IP assay. Precipitates were detected with anti-GFP and anti-FLAG antibodies, respectively. p-SAPK10-FLAG, SAPK10-FLAG phosphorylated band. (**D**) Kinase assay of SAPK10 on ABF1. ABF1 was phosphorylated by SAPK10. Equal amounts of the recombinant proteins were detected with an anti-GST antibody (top panel). The phosphorylated proteins were detected with biotinylated Phos-tag zinc BTL111 complex (bottom panel). p-GST-ABF1, GST-ABF1 phosphorylated band. CIAP, calf intestinal phosphatase. The relative intensity of the GST-ABF1 phosphorylated band was set to 1.00.

**Figure 4 ijms-24-11082-f004:**
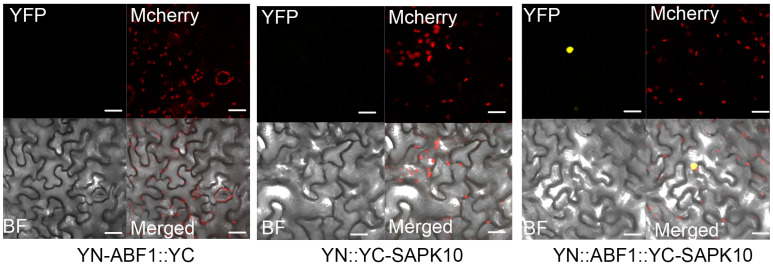
BiFC assay showed the interaction of SAPK10 and ABF1 in tobacco leaf epidermal cells. Red fluorescence represents the auto-fluorescence of chloroplasts, and yellow fluorescence in the nuclei of tobacco leaf cells represents the fluorescence generated by the interaction between ABF1 and SAPK10. Bar = 10 µm.

**Figure 5 ijms-24-11082-f005:**
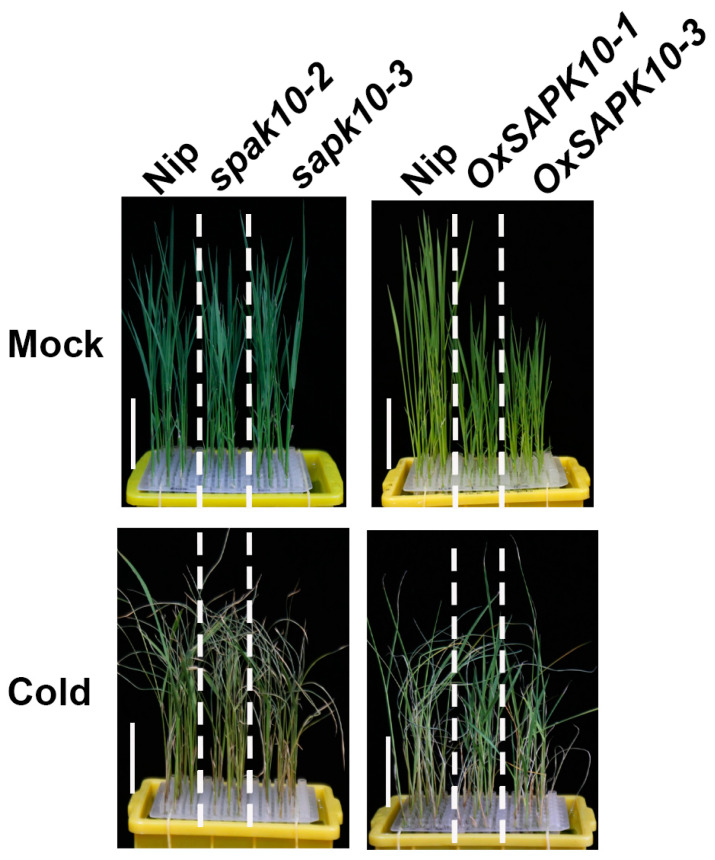
SAPK10 plays a positive role in rice seedling chilling tolerance. *sapk10* mutant lines are less tolerant to chilling stress with more severe wilting and crimping leaves than the wild type. *SAPK10* overexpression lines showed the opposite tendency. Seven-day-old seedlings were hydroponically cultured under chilling stress (4 °C) for 5 days and photographed. Bar = 3 cm.

**Figure 6 ijms-24-11082-f006:**
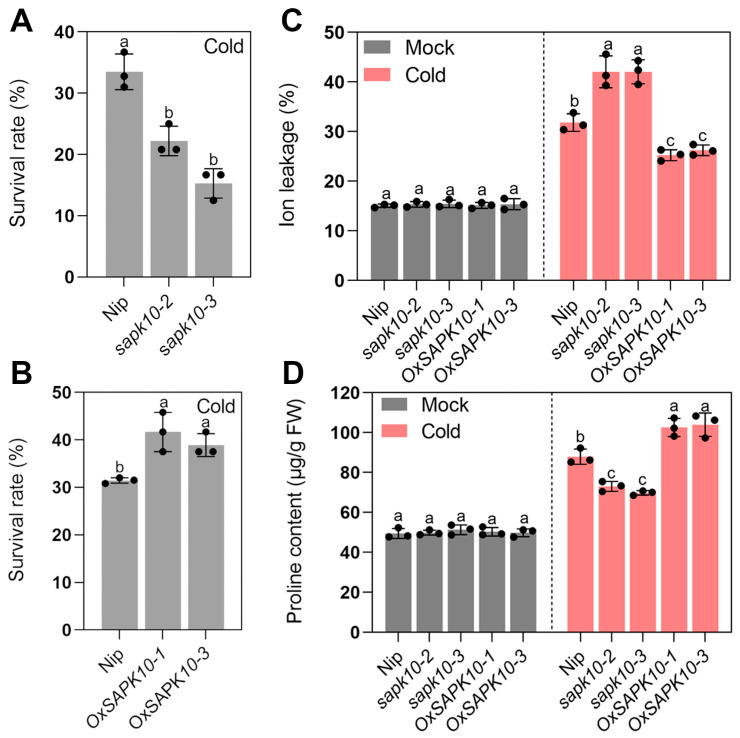
(**A**) Survival rates of wild-type, *sapk10* mutant lines under chilling stress (4 °C) for 5 days. (**B**) Survival rates of wild-type, *SAPK10* overexpression lines under chilling stress (4 °C) for 5 days. (**C**) Ion leakage of seedling leaves from wild type, *sapk10* mutant lines, and *SAPK10* overexpression lines under chilling stress (4 °C) for 5 days. (**D**) Proline contents of seedling leaves from wild type, *sapk10* mutant lines and *SAPK10* overexpression lines under chilling stress (4 °C) for 5 days. Error bars indicate SD with biological triplicates (*n* = 3). Tukey’s test with one-way analysis of variance (ANOVA). Different letters indicate statistical differences at *p* < 0.05.

**Figure 7 ijms-24-11082-f007:**
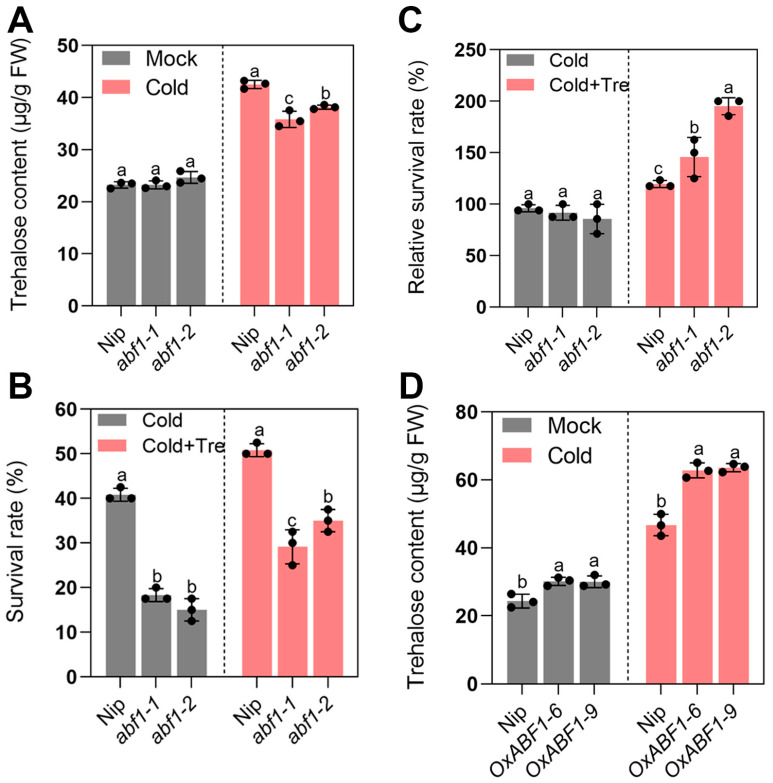
(**A**) Trehalose contents of seedling leaves from wild type and *abf1* mutant lines under chilling stress (4 °C) for 5 days. (**B**) Survival rates of wild-type, *abf1* mutant lines under chilling stress (4 °C) for 5 days. (**C**) Relative survival rates of wild-type, *abf1* mutant lines were expressed as a percentage of those grown under the chilling stress without exogenous trehalose. (**D**) Trehalose contents of seedling leaves from wild type and *ABF1* overexpression lines under chilling stress (4 °C) for 5 days. Error bars indicate SD with biological triplicates (*n* = 3). Tukey’s test with one-way analysis of variance (ANOVA). Different letters indicate statistical differences at *p* < 0.05.

**Figure 8 ijms-24-11082-f008:**
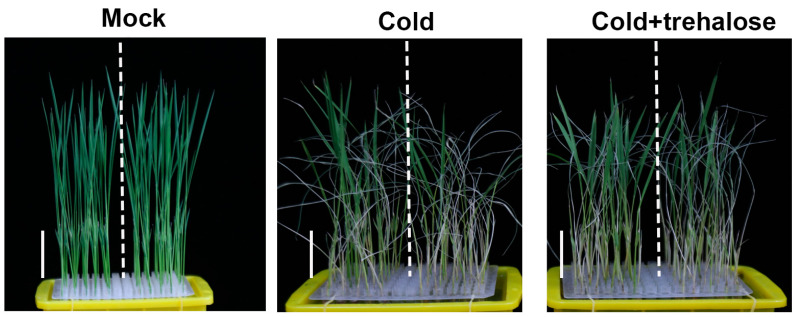
The chilling tolerance of *abf1* mutant lines was enhanced by exogenous trehalose application. Phenotypes of wild type, *abf1* mutant under chilling stress (4 °C) with or without 10 mM exogenous trehalose for 5 days. *abf1* mutant lines are less tolerant to chilling stress with more severe wilting and crimping leaves than the wild type. Applying exogenous trehalose enhanced *abf1* chilling tolerance with less wilting and crimping leaves. Bar = 3 cm.

**Figure 9 ijms-24-11082-f009:**
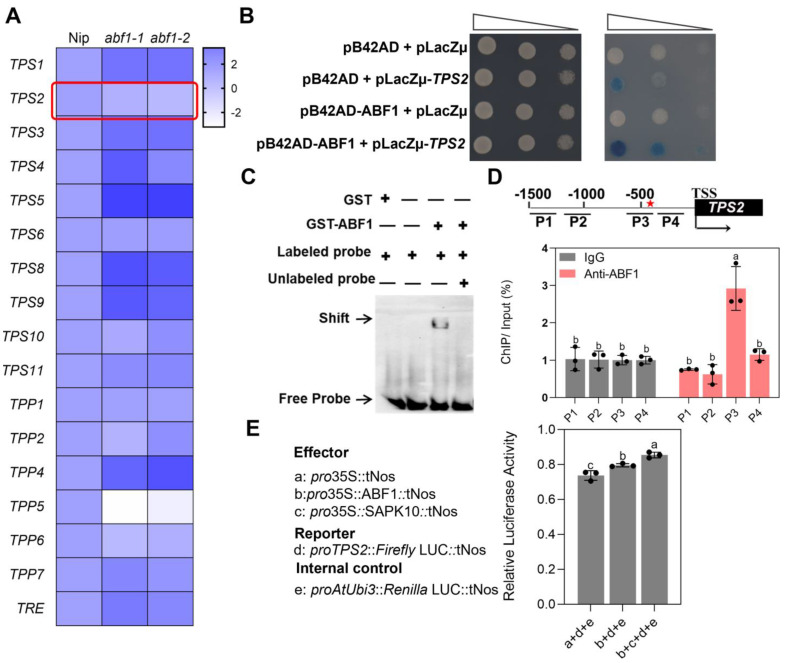
ABF1 directly activates *TPS2* transcription by binding to the G-box in the promoter. (**A**) the expression level of trehalose pathway genes in Nip and *abf1* seedlings under cold stress for 6 h. Data are the means of biological triplicates and were visualized in a heatmap using the log2 fold change of expression ratios relative to Nip. The red box indicated the transcription level of *TPS2* under chilling stress. (**B**) Yeast one-hybrid assay showing the binding of ABF1 to the promoters of *TPS2*. (**C**) Electrophoretic mobility shift assay (EMSA) showing GST-ABF1 specifically binds with the probe 3 (P3) region on the promoter of *TPS2*. (**D**) Probe positions on *TPS2* promoter and genome (**top panel**). Red star: G-box motif (CACGTG) in P3 probe. Black line, untranslated regions; black boxes, exons. Transcription start site (TSS) was set as 0. Numbers indicate the distances (bps) of probes to the TSS. P1–P4, probes 1–4. Chromatin immunoprecipitation quantitative PCR (ChIP-qPCR) assay to show ABF1 binding to the promoter regions of *TPS2* (**bottom panel**). The enrichment values were normalized to the Input. IgG immunoprecipitated DNA was used as a control. Tukey’s test with two-way analysis of variance (ANOVA). (**E**) Luciferase transient transcriptional activity assay in rice protoplast. Effectors: 35S:tNOS, 35S:ABF1:tNOS and 35S:SAPK10:tNOS; Reporter: proTPS2:LUC. Error bars indicate SD with biological triplicates (*n* = 3). Tukey’s test with one-way analysis of variance (ANOVA). In (**D**,**E**), different letters indicate statistical differences at *p* < 0.05.

**Figure 10 ijms-24-11082-f010:**
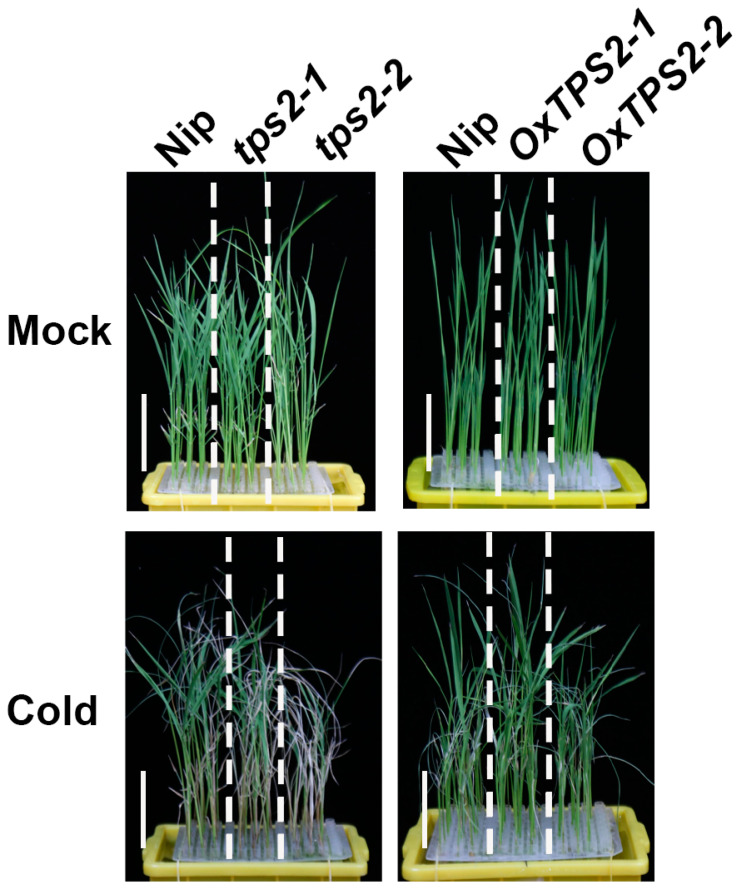
TPS2 enhances rice chilling tolerance. *tps2* mutant lines are less tolerant to chilling stress with more severe wilting and crimping leaves than the wild type. *TPS2* overexpression lines showed the opposite tendency. Seven-day-old seedlings were hydroponically cultured under chilling stress (4 °C) for 5 days and photographed. Bar = 3 cm.

**Figure 11 ijms-24-11082-f011:**
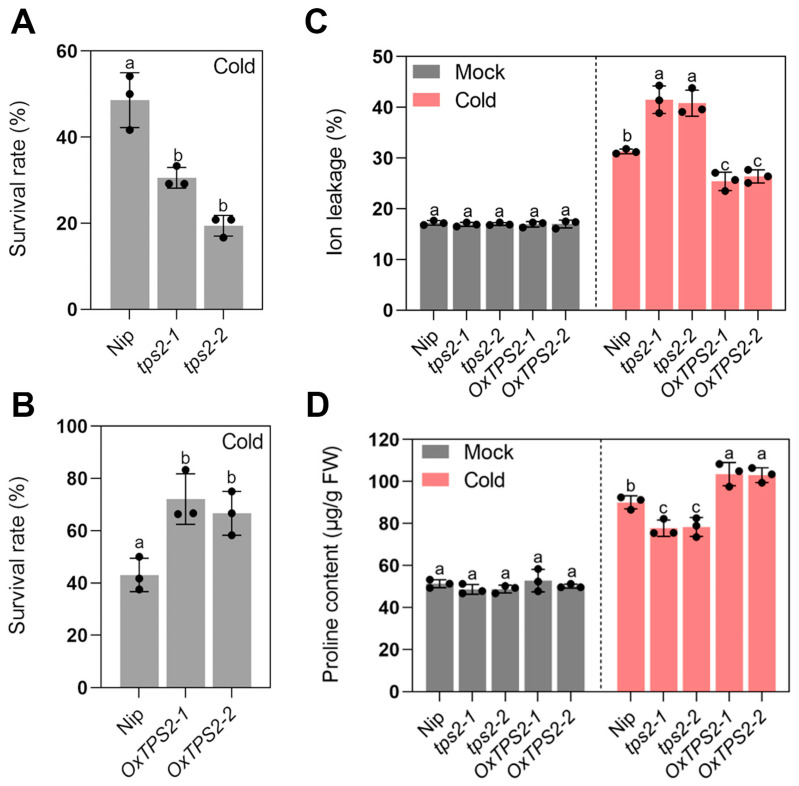
(**A**,**B**) Survival rates of wild type, *tps2* mutant lines and *TPS2* overexpression lines under chilling stress (4 °C) for 5 days. (**C**,**D**) Ion leakage and proline contents of seedling leaves from wild type, *tps2* mutant lines and *TPS2* overexpression lines under chilling stress (4 °C) for 5 days. Error bars indicate SD with biological triplicates (*n* = 3). Tukey’s test with one-way analysis of variance (ANOVA). Different letters indicate statistical differences at *p* < 0.05.

**Figure 12 ijms-24-11082-f012:**
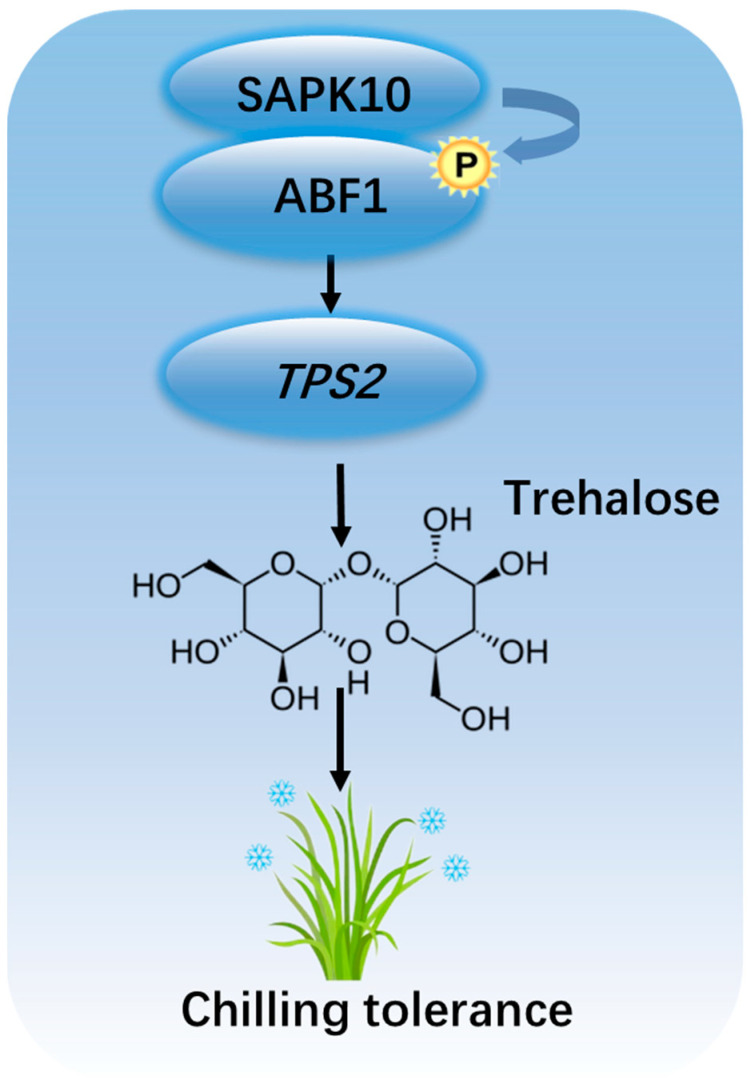
Working model for ‘SAPK10-ABF1-*TPS2*’ pathway in rice chilling tolerance. P, phosphorylation group. Arrowheads show positive regulation, bent arrow represents phosphorylation.

## Data Availability

Data is contained within the article and within Appendix A.

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
