# Peer review of "ABF1 Positively Regulates Rice Chilling Tolerance via Inducing Trehalose Biosynthesis"

_ijms, 2023, doi:10.3390/ijms241311082_

Round 1

Reviewer 1 Report

Dear Authors,

Reviewer comments ijms-2441578

The manuscript entitled „ABF1 positively regulates rice chilling tolerance via inducing trehalose biosynthesis“ represents a useful study aimed at an investigation of a positive role ABF1as a bZIP-type transcription factor on chilling tolerance in rice due to enhancing trehalose 6-phosphate synthase (TPS) transcription level . Moreover, the study also showed that enhanced chilling tolerance can be rescued by exogenous trehalose in tps2-1 and tps2-2 knockout (KO) mutant lines indicating a crucial role of trehalose in enhancement of rice chilling tolerance. The study is focused on the function of three major proteins, ABF1 as a bZIP-type transcription factor, SAPK10, and TPS2 invovlved in trehalose biosynthesis. The function of these proteins was studied in both overexpression and  CRISPR/Cas based knockout (KO) lines of abf1-1, abf1-2, tps2-1, tps2-2, and sapk10-2 and sapk10-3 mutant lines.

I think that the present manuscript is worth publishing in International Journal of Molecular Sciences. I have only a few minor comments on the present manuscript.

Appropriate scale bars in the photos: In the photos of the experimental plant materials and especially in the microphotos of BiFC assay in Figure 2B showing an interaction of SAPK10 and ABF1 in tobacco leaf epidermal cells, appropriate scale bars have to be added to the photographs.

Multidimensional statistical analysis: In addition to the presentation of the individual physiobiochemical characteristics and their comparison in mock- and cold-treated (4 °C for 5 days) overexpression (OX) and knockout (KO) lines and their evaluation by ANOVA and Tukey test at 0.05 level, I think that the authors should provide some kind of multidimensional statistical analysis, e.g., principal component analysis, principal coordinate analysis or cluster analysis in order to obtain novel information on the similarities and the differences in the patterns of the individual characteristics determined to provide a complex view on the relationships between the individual charcteristics determined in the study.

Comprehensive scheme of the novel results: The major item I miss in the present manuscript is an absence of some comprehensive scheme providing a model of SAPK10-ABF1-TPS2 interactions and proposed pathways based on the results of the present study. I think that some scheme providing a model of proposed SAPK10-ABF1-TPS2 interactions based on the results of the present study has to be added to the manuscript.

Formal comments on the text:

Introduction, Line 73: In the reference „Bai et al. (2022)“, an appropriate reference number has to eb given instead of the publication year, i.e., „Bai et al. [reference number]….“

Line 122: Replace the verb „plays“ with „acts“ in the statement „Moeover, ABF1 acts as a negative regulator for flowering transition and is functionally redundant with bZIP40…“

Line 240: Add the word „rice“ following „Indica and Japonica rice.“

Line 252: Add a comma between the words „and“ and „finally“.

Line 254: Modify the statement as follwos: „Hence, a possibility exists that trehalose may play a role in ABF1-mediated low-temeprature tolerance by affecting the active oxygen homeostasis in vivo.“

Line 351: Start a new statement „Based on these regulators which were all induced by cold treatment, the reduced trehalose and proline contents in the mutants under chilling condition…“

Final recommendation: Accept after a minor revision. 

Dear Authors,

Reviewer comments ijms-2441578

The manuscript entitled „ABF1 positively regulates rice chilling tolerance via inducing trehalose biosynthesis“ represents a useful study aimed at an investigation of a positive role ABF1as a bZIP-type transcription factor on chilling tolerance in rice due to enhancing trehalose 6-phosphate synthase (TPS) transcription level . Moreover, the study also showed that enhanced chilling tolerance can be rescued by exogenous trehalose in tps2-1 and tps2-2 knockout (KO) mutant lines indicating a crucial role of trehalose in enhancement of rice chilling tolerance. The study is focused on the function of three major proteins, ABF1 as a bZIP-type transcription factor, SAPK10, and TPS2 invovlved in trehalose biosynthesis. The function of these proteins was studied in both overexpression and  CRISPR/Cas based knockout (KO) lines of abf1-1, abf1-2, tps2-1, tps2-2, and sapk10-2 and sapk10-3 mutant lines.

I think that the present manuscript is worth publishing in International Journal of Molecular Sciences. I have only a few minor comments on the present manuscript.

Appropriate scale bars in the photos: In the photos of the experimental plant materials and especially in the microphotos of BiFC assay in Figure 2B showing an interaction of SAPK10 and ABF1 in tobacco leaf epidermal cells, appropriate scale bars have to be added to the photographs.

Multidimensional statistical analysis: In addition to the presentation of the individual physiobiochemical characteristics and their comparison in mock- and cold-treated (4 °C for 5 days) overexpression (OX) and knockout (KO) lines and their evaluation by ANOVA and Tukey test at 0.05 level, I think that the authors should provide some kind of multidimensional statistical analysis, e.g., principal component analysis, principal coordinate analysis or cluster analysis in order to obtain novel information on the similarities and the differences in the patterns of the individual characteristics determined to provide a complex view on the relationships between the individual charcteristics determined in the study.

Comprehensive scheme of the novel results: The major item I miss in the present manuscript is an absence of some comprehensive scheme providing a model of SAPK10-ABF1-TPS2 interactions and proposed pathways based on the results of the present study. I think that some scheme providing a model of proposed SAPK10-ABF1-TPS2 interactions based on the results of the present study has to be added to the manuscript.

Formal comments on the text:

Introduction, Line 73: In the reference „Bai et al. (2022)“, an appropriate reference number has to eb given instead of the publication year, i.e., „Bai et al. [reference number]….“

Line 122: Replace the verb „plays“ with „acts“ in the statement „Moeover, ABF1 acts as a negative regulator for flowering transition and is functionally redundant with bZIP40…“

Line 240: Add the word „rice“ following „Indica and Japonica rice.“

Line 252: Add a comma between the words „and“ and „finally“.

Line 254: Modify the statement as follwos: „Hence, a possibility exists that trehalose may play a role in ABF1-mediated low-temeprature tolerance by affecting the active oxygen homeostasis in vivo.“

Line 351: Start a new statement „Based on these regulators which were all induced by cold treatment, the reduced trehalose and proline contents in the mutants under chilling condition…“

Final recommendation: Accept after a minor revision.

Reviewer 2 Report

Dear Authors,

I had a great opportunity and honour to review a manuscript/communication entitled: “ABF1 positively regulates rice chilling tolerance via inducing trehalose biosynthesis” which is considered for publication in IJMS. The manuscript focus on new elements of rice chilling tolerance. Manuscript is interesting however need several major improvements also in interpretation of data. The needed improvements are presented in form of list below:

Introduction section:

According IJMS publication rules the Introduction must have a precisely formulated aim of the study currently nothing like that is present. This must be updated.

Results section:

The most problematic section because of Figures. All of them is overloaded with data which made most of it parts too small and too low quality especially in parts with photos. The Figures must be rearranged (split to more Figures).

Figure 1 must be split. The A part must be separate figure a lot of bigger with higher quality. Moreover, The symptoms of increase susceptibility to chill stress must be marked and named in description. I am sorry to say but the Authors did not have any evidence about hypersensitive response of mutants without multilevel analyses of modulation of reactive oxygen species the term Hypersensitive cannot be used.

Figure 2 Must be split. A,C and D must be separate figure than B. The B part must be enlarged and higher quality because fluorescence signal on photos is barely visible.

Figure 3 must be changed. A part must be separate Figure. Again The Authors did not have any evidence about hypersensitive response of mutants without multilevel analyses of modulation of reactive oxygen species the term Hypersensitive cannot be used.

Figure 4 The same problem like in Figure 1. The B part must be separate figure a lot of bigger with higher quality. Moreover, the symptoms of increase reactions to chill stress must be marked and named in description. Phenotypes of plants in B must be characterized in description of Figure.

Figure 6 what exactly is phenotype of overexpression response.? Again split figure.

Sincerely,

The little spelling problems. Some strenge use of term like phenotype in Figure captions

Reviewer 3 Report

there are interesting research, but many questions still remain open.

some points are below:

Line 20: “chilling tolerance ability” = chilling tolerance. Reduced ion leakage and increase in proline reflected rather tolerance??

Line 26: “chilling sensitivity phenotype”??

Lines 31-32: Please, edit: there are two part which cannot connect with “and”.

Line 36: for the breeding physiological, not molecular mechanism has a primary importance.

Line 47: content cannot be induced!

Line 53: H2O2 level cannot be stimulated, only increasing.

Line 59: bZIP52 can not be induced itself, expression can be induced.

Lines 56 – 74: there are nice description of many facts, but text need to be more structured and have more conclusions.

Lines 88- 89: short term chilling did not describe in M&M. Please, add information. In this case dark-light period have a great importance. You can have a different result of chilling stress if you transfer plants after dark period or after prolonged light incubation.

Figure 1, B, C – please, provide information how did you measure survival rate.

Line 120: it is better to write abf1 mutant lines. It is one mutant, but two lines.

Lines 130 – 139: it seems to be the most important points. Chilling stress resistance traits regulated epigenetically with different response in different cell type. It will be nuce to extend this part and in future check regulation of epigenetic in situ.

Line 141: “After screening over one million colonies” – million colonies is redundant!

As the next step, it will be important to demonstrate expression level of AFB1 and SARK10 in “intact” rice plants and shown it co-localization and up regulation in the same cell type.

In plants hormone biosynthesis and distribution are cell position (type) specific and it will be nice further confirmed up regulation of mRNA level in the same cell and corresponding changes of epigenetic status of these cells. This can be the basis of physiological model of chilling stress response.

Fig 3, B, C – Nip data does not fit. Please, explain why.

Figures 1 and 3, D, E should be considered with precaution since different “viability” ratio. How can you exclude that lower Pro and higher ion leakage related with death part of the plants?

Please. Discuss also which cell type/position were affected by chilling.

Lines 227-228: “the abf1 seedlings had 227 lower trehalose levels than the wild-type under cold treatment, while there was no difference under normal condition (Figure 4A)” – how this level related with viability? Maybe in death plants/cells trehalose simple leak out before tissue collection.

Authors have to describe mechanism of chilling recovery: which cell are the primary target of chilling? Photo-oxidative stress in mature leaf? Meristem cell?

Presence of active meristem cell allow plant to recover by formation of new leaf and photo-oxidative stress in mature leaf may lead to carbon starvation, what, in turn, is a key factor of cell viability and activity. Carbon is the structural components of cell wall, DNA, protein etc.

Without carbon supply cell can rapidly died.

 That*s why author’s idea about involvement of trehanose in chilling starvation look very reliable, but required more discussion.

Please, consider role hormones in chilling stress like it was discussed here:

Tiwari, M., Kumar, R., Subramanian, S., Doherty, C. J., & Jagadish, S. K. (2023). Auxin–cytokinin interplay shapes root functionality under low-temperature stress. Trends in Plant Science.

M&M: plant materials descriptions are missing.

Lines 369- 374: please, extend this part: for the low temperature treatment light intensity is a key parameter, but I have not seen it. Do plants have photo- oxidative stress with damage endogenous carbon nutrition?

Line 476: “before or after chilling stress at 4 °C” - this is not correct comparison. Comparison must be done between control and chilling stressed plants of the same age/developmental stage.

moderate corrections

Round 2

Reviewer 2 Report

Dear Authors,

All my suggestions was added. I recomend publication

Sincerely,

Author Response

Again, the authors sincerely appreciate your suggestive comments on our work.

Reviewer 3 Report

Please, read carefully and do minor revision.

Line 102: porcesses ???

Lines 105- 108: please, read carefully and edit. Do not forget that carbon is a main plant element supllying only through photosynthesis!

Lines 363- 369:please, sightly edit to avoid repetitive words combination.

Line 391: "controlling root stem cells"??

ok
